# The Developments of Surface-Functionalized Selenium Nanoparticles and Their Applications in Brain Diseases Therapy

**DOI:** 10.3390/biomimetics8020259

**Published:** 2023-06-15

**Authors:** Rong Hu, Xiao Wang, Lu Han, Xiong Lu

**Affiliations:** 1Key Laboratory of Advanced Technologies of Materials, Ministry of Education, School of Materials Science and Engineering, Southwest Jiaotong University, Chengdu 610031, China; ronghu694914719@163.com (R.H.); eloisewx@163.com (X.W.); 2Key Laboratory of Marine Drugs, Ministry of Education, School of Medicine and Pharmaceutics, Ocean University of China, Qingdao 266003, China

**Keywords:** selenium nanoparticles, polyphenol groups, polysaccharides, surface functionalization, brain diseases

## Abstract

Selenium (Se) and its organic and inorganic compounds in dietary supplements have been found to possess excellent pharmacodynamics and biological responses. However, Se in bulk form generally exhibits low bioavailability and high toxicity. To address these concerns, nanoscale selenium (SeNPs) with different forms, such as nanowires, nanorods, and nanotubes, have been synthesized, which have become increasingly popular in biomedical applications owing to their high bioavailability and bioactivity, and are widely used in oxidative stress-induced cancers, diabetes, and other diseases. However, pure SeNPs still encounter problems when applied in disease therapy because of their poor stability. The surface functionalization strategy has become increasingly popular as it sheds light to overcome these limitations in biomedical applications and further improve the biological activity of SeNPs. This review summarizes synthesis methods and surface functionalization strategies employed for the preparation of SeNPs and highlights their applications in treating brain diseases.

## 1. Introduction

Selenium (Se), a naturally occurring metal-like element [1], was discovered more than 200 years ago by the Swedish chemist Jöns Jakob Berzelius [2]. In nature, Se is contained in the Earth’s crust and is very rare and scattered. Se usually exists in the form of compounds, which can be divided into inorganic and organic forms [3]. Inorganic Se compounds, such as selenium disulfide, selenium sulfide, and selenium dioxide, mainly exist in water and soil, which are difficult to be utilized in the human body because of their high toxicity [4]. In contrast, organic Se compounds are widely found in animals and plants, which are composed of a combination of Se and other organic elements (e.g., carbon, hydrogen, oxygen, and nitrogen) or organic materials (e.g., proteins and amino acids). In general, organic Se mainly exists in the form of selenomethionine that is involved in the metabolic pathway of methionine and participates in synthesis process of proteins [4,5] (Figure 1). The chemical cycle of selenium in living organisms show that plants prefer to absorb Se(IV) and Se(VI) in oxidation states due to their solubility (green arrows in Figure 1). For organic selenium and selenium nanoparticles, they will be oxidized into Se(IV) or Se(VI) for absorption after they enter the organism (solid orange arrows in Figure 1), and then converted into selenoproteins to participate in the metabolic activities of the organism [5]. The selenium intake within the physiological range is essential for the maintenance of various biological functions, including antioxidant defense, redox homeostasis, growth, reproduction, immunity, and thyroid hormone production [6]. The biological effects of selenium are mainly mediated by selenoproteins, which contain at least one selenocysteine (Sec), a selenoamino acid, and most selenoproteins have oxidoreductase activity [7]. Thus, organic Se is easily absorbed and stored by human tissues [5]. However, the toxicity of organic Se remains a major concern for in vivo applications. For example, in the presence of mercaptans, organic Se can be converted to selenols, which will produce reactive oxygen species. In addition, organic selenocysteine can inhibit the methylation of selenium and increase the amount of intermediary metabolite and hydrogen-selenide, which are also toxic [8]. In addition, the dose of inorganic Se should be considered, which limit its bioavailability in vivo.

To address these concerns, nanoscale selenium (SeNPs) with different forms, such as nanowires, nanorods, and nanotubes, have been proposed [8]. Moreover, SeNPs have become increasingly popular in biomedical applications owing to their high bioavailability and bioactivity, and are widely used in oxidative stress-induced cancers, diabetes, and other diseases [9,10]. For example, Fan et al. [11] investigated the antioxidant and protective effects of SeNPs in streptozotocin (STZ)-induced diabetic rats. The results showed that SeNPs were able to mitigate oxidative damage caused by diabetes. In addition to antioxidative ability, SeNPs also possess anticancer potential, and thus can be used as chemoprevention drug. However, pure SeNPs still encounter problems when applied in disease therapy because of their poor stability. Surface functionalization strategy sheds light to overcome these limitations in biomedical applications. With in-depth research, SeNPs have been greatly expanded for both synthesis and application. At the same time, the surface modification of SeNPs also makes the functions more diversified, and its applications are more extensive, such as in tumor and neurodegenerative disease treatment and drug delivery (Figure 2). This review summarizes the synthesis methods of SeNPs and emphasizes the functionalization strategies to improve the bioactivity of SeNPs. The development of SeNPs in the treatment of brain diseases are also reviewed.

## 2. Synthesis Strategy of Selenium Nanoparticles

Se and its organic and inorganic compounds in dietary supplements have been found to possess excellent pharmacodynamics and biological responses. However, the element in its bulk form has low bioavailability and high toxicity, which is tremendously improved upon conversion to the nanoform [16]. Various efficient methods have been reported to synthesize stable SeNPs and their functional derivatives, which generally can be summarized as physical, chemical, and biological methods (Table 1).

### 2.1. Physical Synthesis

Physical synthesis: Commonly used physical methods for the SeNPs synthesis include hydrothermal treatments, microwave irradiation, and laser ablation [39]. For example, Nastulyavichus et al. [40] prepared SeNPs with a high-purity base coating via fast and effective nanosecond laser ablation of the corresponding solids in water, with an average particle size of ~200 nm. Similarly, Gudkov et al. [41] used fiber ytterbium laser and copper vapor laser to ablate the SeNPs in water and adjusted the particle size of SeNPs by changing the laser fragmentation time. In addition, Menazea et al. [17] synthesized polyvinyl alcohol/chitosan-doped SeNPs via one-step laser ablation, which improved the antibacterial activity of the nanoparticles. Although physical synthesis is environmentally friendly, it has problems of high energy consumption, easy contamination of samples, and uneven particle sizes [42]. For example, the laser ablation method requires a Coherent-Vitara laser oscillator to produce fs laser pulses (Figure 3) [43].

### 2.2. Chemical Synthesis

Chemical synthesis: Chemical synthesis of SeNPs involves a redox reaction. The reducing agents, such as ascorbic acid and sodium borohydride (NaBH_4_) are commonly used to cause reduction in the oxidation state, while Na_2_SeO_3_ or selenium dioxide (SeO_2_) are used as Se sources [42]. In terms of the chemical synthesis, the morphology of the SeNPs is also dependent on the reducing agents. For example, Chandramohan et al. [44] prepared SeNPs with different morphology (rod, spherical, and square) using different reducing agents, bovine serum albumin, D-glucose, and soluble starch, respectively. The SeNPs with different morphologies exhibit different antibacterial and antioxidant properties. However, bare SeNPs have poor dispersibility and is easily oxidized. To solve the problem of chemically synthesized SeNPs easily agglomerating and their inability to exist in a stable state for a long time, surfactants, such as polyvinyl alcohol (PVA), polyvinylpyrrolidone (PVP), and polyethylene glycol (PEG) are employed. For example, Cao et al. [18] synthesized water-soluble and stable SeNPs from gray selenium using PEG as surfactants (Figure 4). The results showed that the synthesized SeNPs was dispersed spherically and had good stability. In short, the SeNPs with different morphology can be easily obtained from the chemical synthesis by changing the kinds of reducing agents. However, the chemical synthesis may involve toxic chemicals.

### 2.3. Biosynthesis Strategy

Biosynthesis strategy: Biosynthesis, also known as green synthesis, rely on plants [46], fungi [47], and bacteria to synthesize SeNPs, which possesses the advantages of being environmentally friendly, less toxic, and economical compared to other methods. For example, Xu et al. [46] synthesized SeNPs by fermentation under anaerobic conditions using the probiotic Lactobacillus casei ATCC 393 as the starting strain and Na_2_SeO_3_ as the selenium source (Figure 5). In another study, Gonzalez-Salitre et al. [48] investigated the progress of bioproduction of Se nanoparticles by probiotic yeast due to their ease of culture, lack of toxicity and pathogenicity. Fan et al. [11] realized the green synthesis of SeNPs by using rose eggplant leaf extract as the reducing agent. In another study, Miglani et al. [49] extracted SeNPs from fresh guava leaves. These studies show that biosynthesis method is not only safe and efficient but also results in SeNPs with better physical and chemical properties.

## 3. Surface Functionalization of Selenium Nanoparticles with Enhanced Bioactivity

The SeNPs can be functionalized by integrating a variety of drugs or biomacromolecules based on covalent or non-covalent coupling, such as the chemical conjugation of biomacromolecules on SeNPs and the physical adsorption onto the surface SeNPs [50]. To achieve better disease treatment, signal regulation, and targeting, researchers modified and functionalized the surface of SeNPs by some functional groups with special properties (such as polyphenols [23,51,52], polysaccharides with special functional groups [24,53,54,55,56,57,58], proteins [59], etc.), therapeutic drugs (doxorubicin [60], irinotecan [38], etc.), and target genes [22].

### 3.1. Polyphenols-Functionalized Selenium Nanoparticles

Natural polyphenols are a class of natural molecules with two or more phenolic hydroxyl groups that are widely present in fruits, vegetables, tea, plant seeds, and Chinese herbal medicine [61]. The strong antioxidant capacity of natural polyphenols has been widely exploited in the biomedical field. In particular, polyphenols can form dynamic covalent interactions or strong non-covalent interactions between catechol/pyrogallol with other functional groups, which therefore can be easily hybridized with a variety of building groups to form multifunctional composites, further broadening their applications [19,20,21,61,62]. The combination of SeNPs and polyphenols can synergistically enhance the antioxidant properties of SeNPs and endow SeNPs with properties inherent to natural polyphenols, such as adhesion. It was confirmed that the stability and oral availability of epigallocatechin gallate modified SeNPs were improved by ascorbic acid reduction [63]. Wang et al. [25] used gallic acid (GA) to reduce and modify SeNPs, which not only overcame the shortcomings of easy oxidation of GA and instability of Na_2_SeO_3_, but also obtained SeNPs with improved broad-spectrum antibacterial activity. Kumari et al. [64] prepared curcumin-loaded SeNPs (Cur@SeNPs) with uniform size and anticancer effect (Figure 6). In another study, the resveratrol modification of SeNPs not only endowed the SeNPs with anti-oxidative activity, but also improved the binding affinity of SeNPs to amyloid-β (Aβ), which had great potential in neuroprotection [65].

### 3.2. Polysaccharide-Functionalized Selenium Nanoparticles

Polysaccharides are the cheapest and most abundant biopolymers in the biosphere, and are formed by the natural polymerization of monosaccharides through different glycosidic bond types. Compared with other biopolymers, they are rich in reactive functional groups, such as –OH, –COOH, –NH_2_ and –OSO_3_, with excellent adjustable performance and different physical, chemical, and biological characteristics [19,26,27]. Polysaccharide-functionalized SeNPs also have many advantages, such as high biocompatibility, biodegradability, and active hydroxyl groups [28]. For example, SeNPs functionalized with chitosan (CS), exhibited both enhanced antioxidant and antimicrobial properties [29,56]. The SeNPs were also coupled with hyaluronic acid (HA) to form tumor targeting vector HA@SeNPs, which were then used to encapsulate paclitaxel [51]. In addition, Lentinan was used to functionalize SeNPs, which can target the mitochondria of tumor cells and induce their apoptosis of tumor cells [24]. Wang et al. [30] combined SeNPs with chestnut polysaccharide and significantly improved the antioxidant activity of chestnut polysaccharide. Yang et al. [34] constructed stable and size-controlled lichenan-modified SeNPs by chemical reduction, which also exhibited strong antioxidant activity (Figure 7A). Rao et al. [31] developed a traditional Chinese medicine, astragalus polysaccharide (APS)-decorated SeNPs to deliver tanshinone IIA (TSIIA) (TSIIA@SeNPs-APS), which displayed enhanced antioxidant activity compared to pure SeNPs (Figure 7B). At the same time, the results also showed that TSIIA@SeNPs-APS played a critical role in regulating selenoprotein for the treatment of spinal cord injury. Chen et al. [35] prepared SeNPs in a simple redox system using polygonatum polysaccharide (PSP) as a stabilizer. The PSP@SeNPs exhibited better protective behavior against PC-12 cell activity induced by H_2_O_2_ than pure SeNPs. Zhou et al. [32] successfully fabricated dextran-functionalized SeNPs, which exhibited good anticancer activity in vivo or in vitro. In addition, Tang et al. [33] prepared SeNPs using larch arabinogalactose as a stabilizer and explained the anti-tumor activity of SeNPs based on inhibiting tumor cell proliferation and internalization by tumor cells through endocytosis, which induces tumor cell apoptosis. Furthermore, multifunctional SeNPs can be achieved by the polysaccharide–polyphenol synergy have better stability, stronger antioxidant properties, and anticancer effects than the single modified SeNPs [52].

### 3.3. Selenium Nanoparticles Modified by Other Functional Materials

In addition to the use of polyphenols and polysaccharides to functionalize SeNPs, other functional materials have been employed to endow SeNPs with more specific properties (e.g., proteins [36], peptides [37], and drugs). For example, human serum albumin (HSA)-coated SeNPs can target mitochondria [67]. Peptide-modified SeNPs can target inflammation [68]. Polysaccharide-protein complexes wrap SeNPs, which have stronger anticancer activity [69] and can effectively promote bone formation in vitro and in vivo [70]. Deng et al. [71] prepared SeNPs using solvent diffusion/in situ reduction and loaded with the hypoglycemic drugs Mulberry leaf and Pueraria Lobata for synergistic treatment of diabetes. Xia et al. [72] also employed galactose-modified SeNPs as tumor targeting components to prepare tumor targeted delivery vector and then loaded doxorubicin onto the surface of nanomaterials to improve the anti-tumor effect of doxorubicin in the treatment of liver cancer. In addition, SeNPs can also be used as a gene vector. For instance, Xia et al. [22] prepared functional SeNPs as a non-viral tumor-targeting vector (RGDfC-SeNPs). The RGDfC (Arg-Gly-Asp-DPhe-Cys) could specifically bind to the overexpressed α_v_β_3_ integrin in tumor cells, and the positively charged RGDfC could also promote ligation between nanoparticles and siRNA through their electrostatic interactions. Thus, the RGDfC-modified SeNPs was used to selectively deliver siRNA to HepG2 liver cancer cells and tissues for the treatment of hepatocellular carcinoma (Figure 8). In addition to the examples above, there are numerous other similar studies, and these SeNPs with unique capabilities offer further therapeutic options for a variety of illnesses.

In short, biomolecules, such as polysaccharides and polyphenols, can be used to not only stabilize the SeNPs due to their abundant hydroxyl, carbonyl, and other functional groups that interact with SeNPs, but also can be used to enhance the bioactivities of functionalized SeNPs with the synergistic effects derived from the biomolecules themselves.

## 4. The Biomedical Applications of Functionalized Selenium Nanoparticles

Compared to inorganic Se and organic Se, SeNPs have higher biosafety and bioavailability, and the functionalization of SeNPs can affect their toxicity, surface electrical properties, and stability [8]. Functionalization of SeNPs can also enhance their antibacterial, antioxidant, and anticancer activities. It also provides the properties of the functionalized substance, such as targeting and signal regulation. Therefore, functionalized SeNPs have been widely used for the treatment of cancer, diabetes, and neurological diseases. This section focuses on the latest research progress on functionalized SeNPs for the treatment of brain diseases.

### 4.1. Neurodegenerative Diseases

Neurological disorders are the second leading cause of death and the leading cause of disability worldwide. In central nervous system (CNS) diseases, almost no drug treatment can achieve complete neurovascular recovery; it can only slow the process of neurodegeneration. This situation highlights how difficult it is for current pharmacological drugs to target and function effectively in the brain. One of the major obstacles is the blood–brain barrier (BBB), along with other factors that must be considered, such as the existence of other extracellular and intracellular barriers, complexity of neurovascular networks, and interactions at several level [73]. Decades of research has identified the genetic factors and biochemical pathways involved in neurodegenerative diseases, of which, eight features have been supported by evidence: pathological protein aggregation, synaptic and neural network dysfunction, abnormal protein balance, abnormal cytoskeleton, changes in energy homeostasis, DNA and RNA defects, inflammation and neuronal death [74].

Inflammation and oxidative stress are the two main aspects we focus on when studying SeNPs for the treatment of neurodegenerative diseases. Reactive oxygen species (ROS) are a class of highly reactive molecules, including superoxide anion (O^2−^), hydrogen peroxide (H_2_O_2_), and hydroxyl radical (OH), which are the natural by-products of aerobic metabolism initially generated by oxygen reduction [75,76,77]. Overproduction of ROS induces oxidative stress, a deleterious process that can cause severe oxidative damage to cellular structures. With the aid of various antioxidant enzymes produced in the organism, such as superoxide dismutase and catalase, this redox imbalance can be rapidly restored to the normal state and the redox homeostasis can be balanced by scavenging excess ROS [75].

In addition, many selenoenzymes with redox activity, such as glutathione peroxidase (GPx), and thioredoxin reductase (TrxR), are not only involved in antioxidant defense and cellular redox signaling regulation, but may also be involved in a variety of basic functions of the central nervous system (CNS) [78]. Selenoproteins, for example, play an indispensable role in neuroprotection and neuronal cell survival by scavenging ROS, regulating intracellular calcium and anti-inflammatory effects. At the same time, a large body of evidence suggests that selenoproteins also play a decisive role in normal behavioral development, mood and cognition, and other aspects of psychological functioning, such as depression and anxiety [79]. GPx1 is a ROS scavenger that is expressed in both neurons and astrocytes and plays a key regulatory role in protecting nerve cells from oxidative stress, such as dopaminergic neurons [79]. Therefore, SeNPs, as an emerging form of pure selenium, have high biological activity and free radical scavenging ability. After entering the human body, SeNPs can be absorbed and utilized; participate in the metabolism, transformation, and regulation of selenoenzymes; and show strong therapeutic potential in the treatment of neurological diseases [80].

#### 4.1.1. Alzheimer’s Disease

Alzheimer’s disease (AD) is a common and destructive disease characterized by the aggregation of Aβ peptides. Amyloidosis is a proteotoxic disease that affects the nervous system, and dementia is the most common form of AD [81]. Controlled release, targeted, and multi-channel therapy to inhibit Aβ aggregation is considered key to the success of AD drugs. For example, Gong et al. [80] coated PDA and borneol on the surface of SeNPs through the self-assembly and amidation reaction under alkaline conditions, which enhanced the anti-oxidation and BBB penetrating ability of SeNPs (Figure 9). In vivo and in vitro studies demonstrated that Se@PDA@Bor had multi-enzyme activity, which could remove ROS and active nitrogen (RNS), penetrate the BBB, inhibit the activation of microglia and astrocytes, reduce the release of pro-cellular inflammatory factors, alleviate neuroinflammation, break Aβ aggregation, and rescue the memory and behavior disorders of AD mice.

Huo et al. [82] constructed the Se nanosphere drug delivery system for loading curcumin and poly(lactide-glycolide). Studies have shown that the Se nanosphere could reduce the load of Aβ in the brain of AD mice, significantly improve the memory impairment of AD mice, and inhibit Aβ plaque aggregation and inflammation, which is expected to become an alternative drug for the treatment of AD.

Combining resveratrol (Res) and SeNPs can improve the solubility and bioavailability of resveratrol and enhance its antioxidant activity. For instance, Abozaid et al. [83] used resveratrol to synthesize functional SeNPs in situ. The results showed that the resveratrol functionalized SeNPs could alleviate neurotoxicity, inhibit the production of Aβ, and regulate various signaling pathways involved in the pathogenesis of AD, which is expected to become another alternative drug for the treatment of AD. Similarly, Li et al. [84] decorated SeNPs with resveratrol and loaded the BBB transport peptide (TGN) on the surface to prepare the nanocomposite TGN-Res@SeNPs (Figure 10). The results showed that the TGN-Res@SeNPs could effectively inhibit the deposition of Aβ in the hippocampus, reduce oxidative stress, inhibit neuroinflammation in vivo, and alleviate intestinal microflora disorders to improve cognitive impairment, so as to achieve the purpose of treatment of AD. Conversely, Sun et al. [85] designed mesoporous SeNPs to load resveratrol, β-cyclodextrin nano-valve, and borneol. The experimental results showed that the SeNPs release system could release borneol and open BBB to reduce oxidative stress and inhibit the accumulation of Aβ through the release of resveratrol, and also improve the memory impairment of experimental mice, which demonstrated that the mesoporous SeNPs could be used as potential nanocarrier for the treatment of AD and other neurodegenerative diseases.

Epigallocatechin gallate (EGCG) can also partially protect cells from Aβ-mediated neurotoxicity by inhibiting Aβ aggregation. For example, Zhang et al. [86] prepared EGCG@SeNPs for encapsulating Tet-1 neuropeptide through an electrostatic bonding effect. The results showed that Tet-1-EGCG@SeNPs could inhibit Aβ aggregation and remodel the Aβ into non-β-folded spherical aggregates, which could improve the efficiency of nerve targeting in vitro and greatly improve the efficacy of drugs, thus laying the foundation for the treatment of AD. In another study, Yang et al. [87] synthesized in situ CS/DMY@SeNPs with surface modification of dihydromyricetin (DMY) and chitosan (CS) for loading BBB targeting transporter peptides (Tg). The results suggested that Tg-CS/DMY@SeNPs could penetrate the BBB and inhibit Aβ aggregation and inflammation, and thus it might serve as an ideal candidate for the treatment of AD (Figure 11).

In addition to the combination with polyphenols, sometimes the direct modification of SeNPs with polysaccharides/peptides can reduce neurotoxicity, inhibit Aβ aggregation, and prevent AD. For example, Yang et al. [88] first synthesized SeNPs via the redox reaction, and then coupled the BBB transport peptide (TGN) and short peptide LPFFD to form bifunctional SeNPs (L1T1-SeNPs). The results indicated that dual-function SeNPs could effectively inhibit Aβ aggregation through the BBB, which was a promising approach for the treatment of AD. Gao et al. [89] synthesized chondroitin sulfate functionalized SeNPs, which could reduce oxidative stress damage, inhibit excessive tau phosphorylation, reduce inflammation, and consequently delayed the development of AD.

#### 4.1.2. Parkinson’s Disease

Parkinson’s disease (PD), the second progressive neurodegenerative disease after AD, is pathologically characterized by the loss of dopaminergic neurons in the substantia nigra and the presence of protein inclusion bodies called Lewy bodies. The pathogenesis of PD remains unclear, however, various studies have shown that oxidative stress is an important pathological tool in PD, leading to neuronal death and apoptosis. Therefore, reducing oxidative stress may be a promising strategy for the prevention and treatment of behavioral abnormalities in PD [90,91,92]. Although there have been a few reports on the treatment of PD with SeNPs, some studies are still underway. For example, Yue et al. [91] studied the neuroprotective effects of glycine SeNPs (Gly@SeNPs) in PD rats (Figure 12). The results suggested that Gly@SeNPs could improve the oxidative stress and behavior disorder of PD rats and reduce the loss of dopaminergic neurons, which had the potential to be used as a therapeutic drug for neurodegenerative diseases, such as PD.

#### 4.1.3. Huntington Disease

Huntington disease (HD) is an autosomal dominant neurodegenerative disorder that causes brain cell death. Adult HD patients present motor, cognitive, and psychiatric disorders [93]. Inhibiting the aggregation of Huntington’s protein, promoting neuronal survival, and alleviating oxidative stress may be the key to the treatment of HD [94]. Cong et al. [94] prepared size-stable SeNPs by chemical reduction using BSA as the stabilizer and investigated its therapeutic effect in an HD model (Figure 13). The results showed that the SeNPs could attenuate oxidative stress, inhibit HD protein aggregation, and down-regulate the expression of histone deacetylase family members, which had great potential in HD treatment.

#### 4.1.4. Depression

Depression is one of the most common neuropsychiatric disorders, affecting approximately 300 million people worldwide [95]. Sufficient evidence suggests that neurooxidative damage, inflammation, and apoptosis associated with neurodegeneration, reduced neurogenesis, and reduced neuroplasticity are major mechanisms underlying the onset of depression [96]. Based on this, Albrakati et al. [97] used prodigiosins (PDGs) to reduce and modify SeNPs, and investigated its effect on depression-like behavior induced by chronic and unpredictable mild stress (CUMS) in rats. The results suggested that PDGs@SeNPs exerted neuroprotective effect on CUMS rats by preventing neuroinflammation, ameliorating oxidative stress, preventing neuronal cell loss, and activating glial fibrillary acidic protein in the hippocampus. Therefore, PDGs@SeNPs might be a potential antidepressant candidate due to their excellent antioxidant, anti-inflammatory and neuroprotective performances. Yang et al. [98] also prepared SeNPs based on the template method and explored its effect on sodium fluoride (NaF)-induced depression-like behaviors (Figure 14). The results also demonstrated that the SeNPs could reduce the branching index of microglia cell, inhibit NaF-induced neuroinflammation, restore dopamine and norepinephrine secretion, and alleviate depression by inhibiting the JAK2-STAT3 pathway.

#### 4.1.5. Stroke

Stroke is defined as a neurological dysfunction resulting from acute focal damage to the central nervous system (i.e., brain, retina, or spinal cord) due to vascular causes [99]. A major challenge in the development of anti-stroke drugs is the complexity of the signaling processes and the associated inflammatory response. Based on the protective effect of the SeNPs in the CNS, Amani et al. [100] fabricated OX26-PEG@SeNPs by coating SeNPs with polyethylene glycol (PEG) and loading anti-transferrin receptor mab (OX26) into the SeNPs. It was found that OX26-PEG@SeNPs could enhance the targeted transport of SeNPs into the brain, target different cell signaling pathways to regulate cell metabolic state and inflammation, enhance the functional properties of hippocampal neurons, and thus promote neuron survival for stroke treatment (Figure 15). Lv et al. [101] encapsulated gallic acid (GA) into SeNPs to obtain GA@SeNPs, which provided a therapeutic strategy for ischemic stroke based on SeNPs-induced inhibition of excessive inflammation and oxidative metabolism.

### 4.2. Brain Glioma

Glioma is the most insidious and destructive brain tumor and the fourth leading cause of cancer-related death. All available therapies face the profound challenge of being unable to penetrate the BBB and the blood–brain tumor barrier (BBTB) to deliver systemic chemotherapy at the tumor site. Based on this, Song et al. [102] prepared an “intelligent” nano-system based on SeNPs-loaded HER2 antibodies that were modified by therapeutic agents and ultrasonic diagnostic agents. The results showed that the nano-system enhanced the BBB permeability of SeNPs by HER2 antibody, and enhanced brain tumor therapy and magnetic resonance imaging, while SeNPs inhibited the growth of tumor globules by triggering the p53 signal pathway mediated by DNA damage (Figure 16). This research provided a cancer-targeting nano-system that could overcome the BBB and hold promise for precision treatment of human malignant gliomas. In another study, Huang et al. [103] synthesized DOX-encapsulated SeNPs and loaded RGD and c-myc-siRNA into the SeNPs. Their results also suggested that the SeNPs could efficiently penetrate the BBB and achieve synergistic treatment of glioblastoma by triggering intracellular ROS overproduction and releasing DOX under acidic conditions. Chen et al. [104] designed bamboo-dervied polysaccharides functionalized SeNPs as a tumor targeting nano-system to combat BBB, thus enhancing its anti-glioma ability. To sum up, functionalized SeNPs can successfully penetrate the blood–brain barrier, providing a strong therapeutic basis for overcoming neurological disorders.

## 5. Summary and Prospects

Selenium is an essential trace element that participates in the regulation of many signal mechanisms in the human body. Selenium deficiency will lead to immune dysfunction, and even cause some cardiovascular diseases, while high-dose selenium intake will lead to selenium poisoning. With controllable size, high activity, low toxicity and easy chemical modification, SeNPs with high specific surface area and low excretion rate have shown high uptake efficiency, transformation efficiency and good therapeutic effect in the treatment of neurological diseases. The bioactivity of SeNPs depended on their size due to the different endocytosis pathways induced by different-sized SeNPs. Recent studies focused on the effects of SeNPs on neuroinflammation or cancer cells. It is also important to investigate the effect of SeNPs on normal cells in terms of tissue repairing fields. In addition, surface-functionalized SeNPs with polyphenols and polysaccharides can be used as both nanodrugs and nanocarriers for the treatment of neurological diseases can successfully penetrate the blood–brain barrier, target different cell signaling pathways, improve the oxidative stress state of cells, and regulate neuroinflammation, thereby promoting the protection of nerve cells and treating neurodegenerative diseases. In conclusion, functionalized SeNPs have a non-negligible potential for the treatment of brain diseases. However, the synergy mechanism of the SeNPs with polyphenols and polysaccharide in treating diseases requires in-depth exploration in the future. It is believed that in the near future, given the beneficial effects of selenium on the human body and some other biological characteristics, more and more researchers will develop and design selenium nanomaterials and their derivatives to be used in other biomedical fields, such as nutritional supplements and biosensing.

## Figures and Tables

**Figure 1 biomimetics-08-00259-f001:**
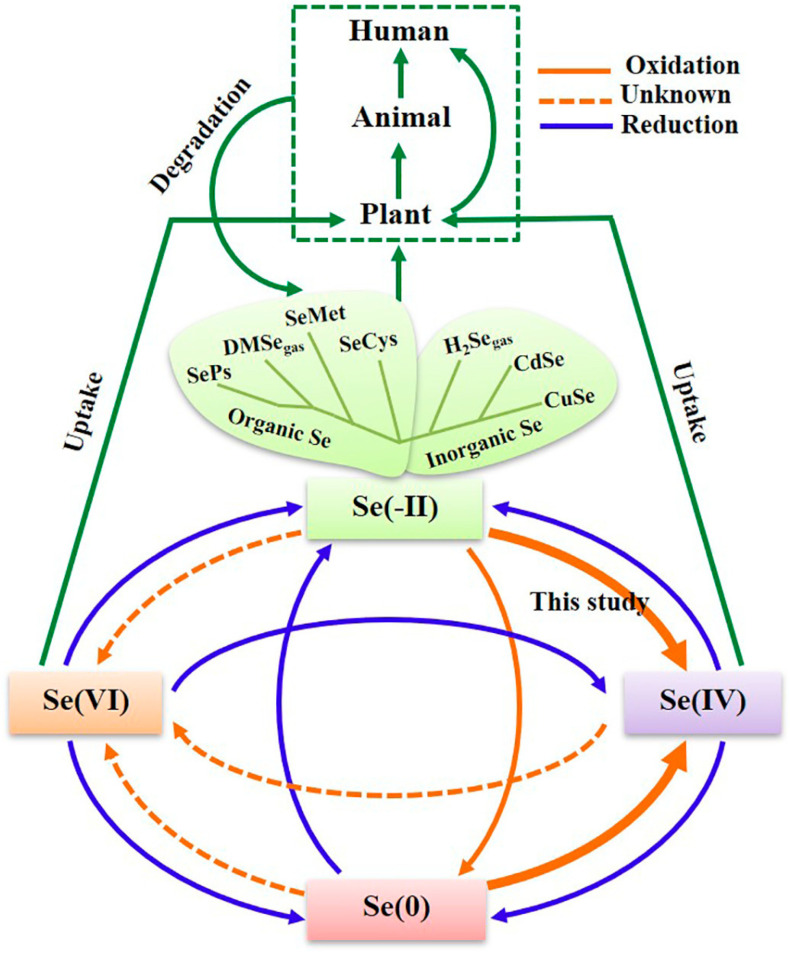
The proposed biogeochemical cycle of Se. Reprinted from [5] with permission from Elsevier.

**Figure 2 biomimetics-08-00259-f002:**
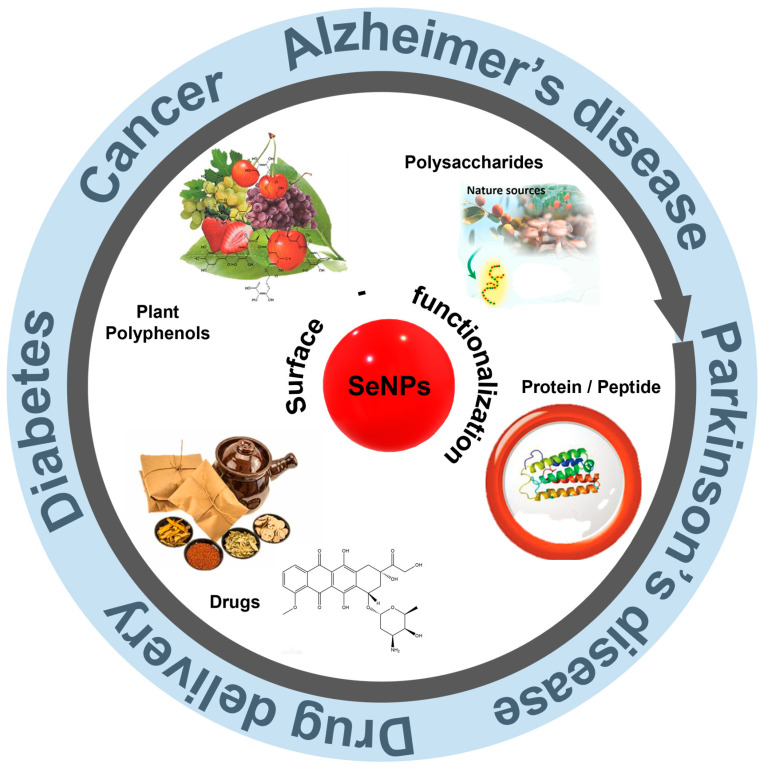
A schematic diagram of surface functionalization and therapeutic application of SeNPs. Reprinted from [12,13] with permission from Elsevier. Reprinted from [14,15] with permission from John Wiley and Sons.

**Figure 3 biomimetics-08-00259-f003:**
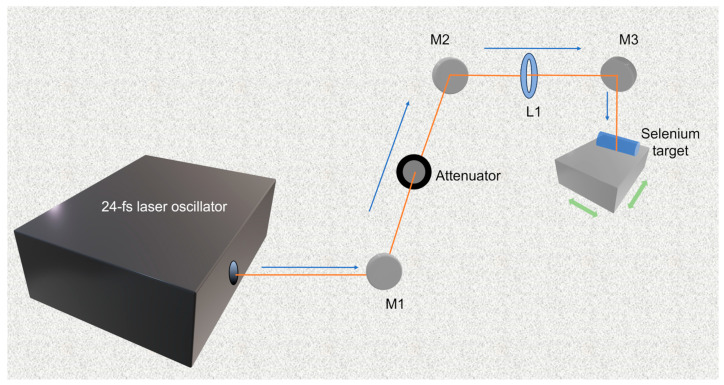
Schematic illustration of synthesizing process of SeNPs via femtosecond pulsed laser ablation in liquids. Reprinted from [43] with permission from Springer Nature.

**Figure 4 biomimetics-08-00259-f004:**
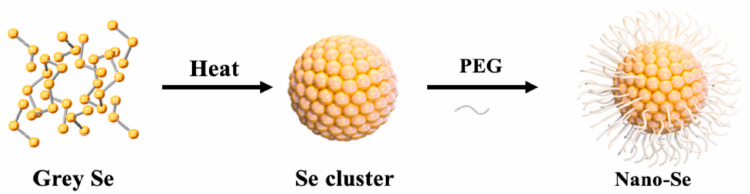
Schematic illustration synthesis of PEG-stabilized SeNPs. Reprinted with permission from [18] published under the Creative Commons Attribution license (CC BY-NC-ND 4.0) [45].

**Figure 5 biomimetics-08-00259-f005:**
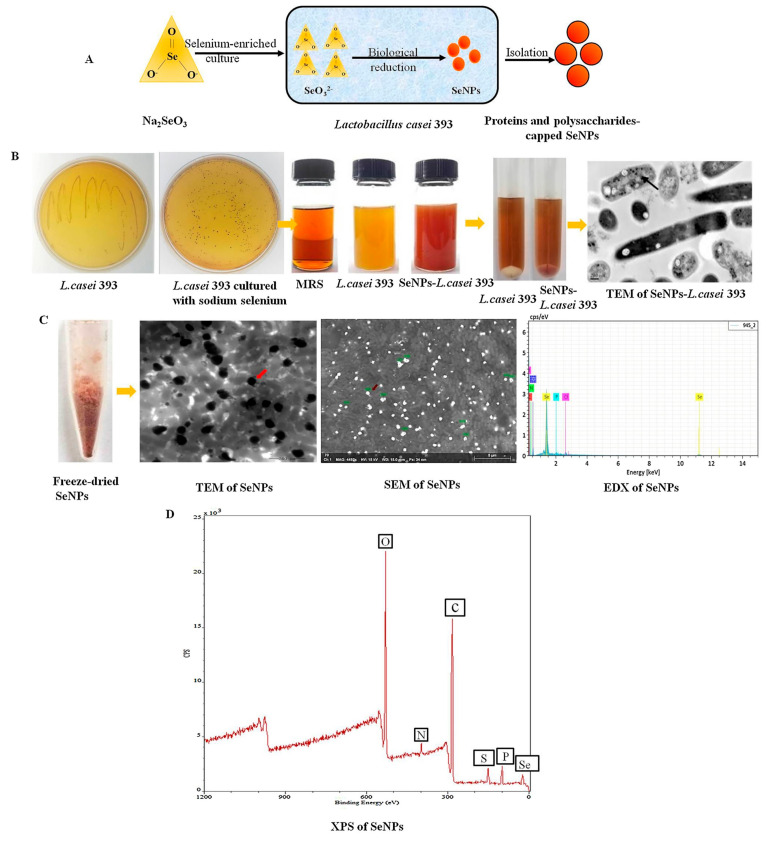
(**A**) Schematic representation of the hypothesis of SeNPs synthesis by the probiotic Lactobacillus casei 393. (**B**) Macroscopic schematic of SeNPs biosynthesis, and TEM images. (**C**) Extracted biogenic biomolecules capped-SeNPs powder appeared dark red after freeze-drying. Moreover, these SeNPs were with the size of 50–80 nm particles analyzed via scanning electron microscopy (SEM) and TEM, and composition of carbon (C), nitrogen (N), oxygen (O), selenium (Se) and phosphorus (P) elements were analyzed via energy dispersive X-ray spectrometry (EDX). (**D**) Extracted biogenic biomolecules capped-SeNPs contained C, N, O, Se and P elements via X-ray photoelectron spectroscopy (XPS) analysis. Reprinted from [46] with permission from Elsevier.

**Figure 6 biomimetics-08-00259-f006:**
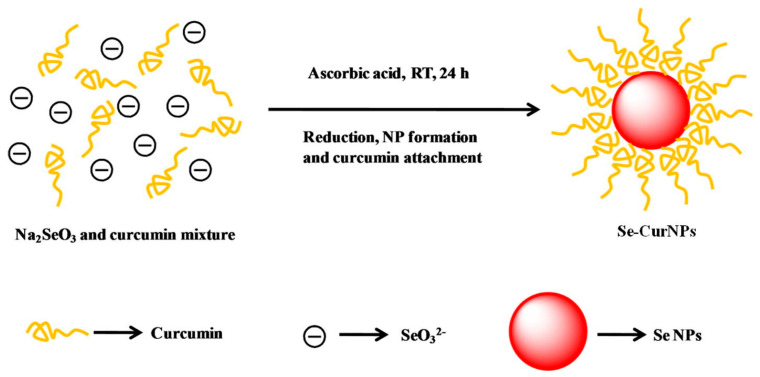
Schematic representation for the synthesis of Cur@SeNPs. Reprinted from [64] with permission from Elsevier.

**Figure 7 biomimetics-08-00259-f007:**
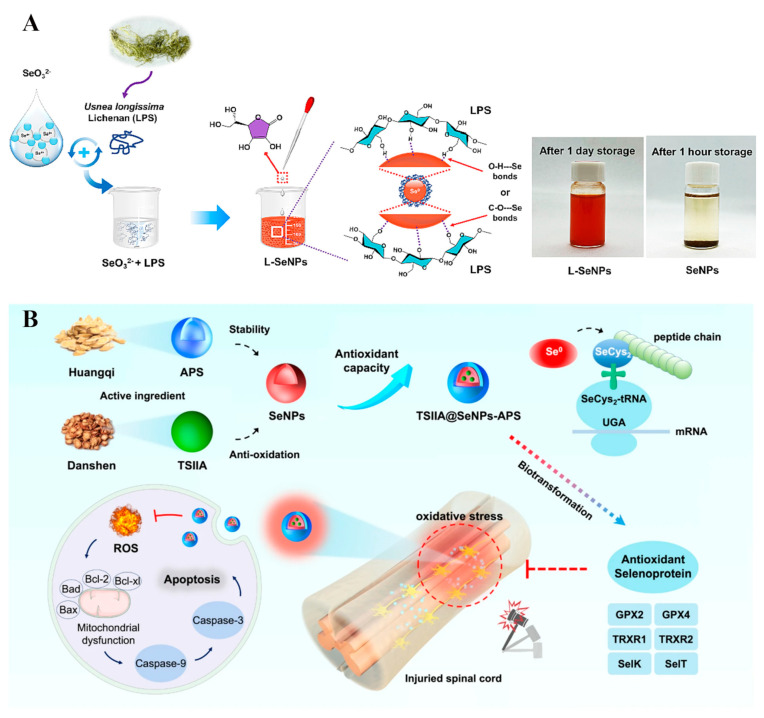
(**A**) Schematic diagram of synthesis of lichenan-modified SeNPs. Reprinted from [34] with permission from Elsevier. (**B**) Synthetic schematic diagram of TSIIA@SeNPs-APS and its regulation of antioxidant selenoproteins for SCI treatment. Reprinted with permission from [31] published under the Creative Commons Attribution license (CC BY 4.0) [66].

**Figure 8 biomimetics-08-00259-f008:**
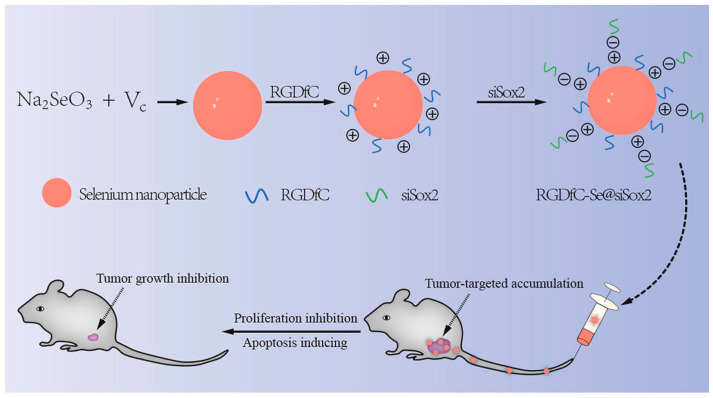
Schematic representation of functional SeNPs delivering siRNA to HepG2 for hepatocellular carcinoma treatment. Reprinted with permission from [22] published under the Creative Commons Attribution license (CC BY-NC-ND 4.0) [45].

**Figure 9 biomimetics-08-00259-f009:**
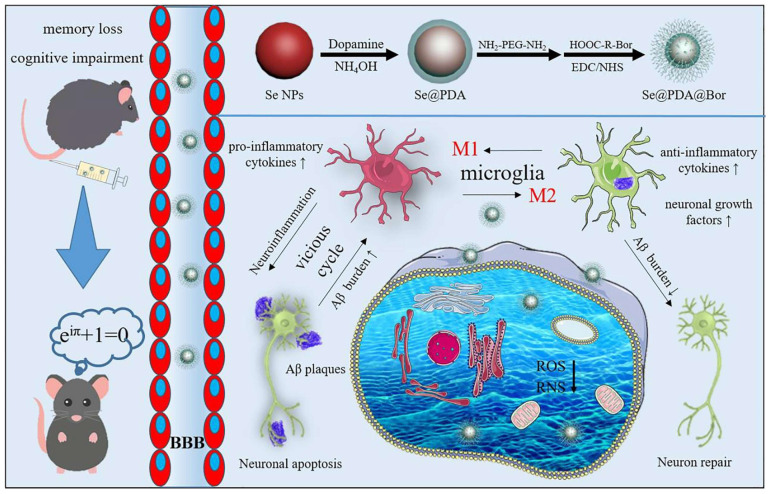
Schematic diagram of the function of Se@PDA@Bor. Reprinted from [80] with permission from Elsevier.

**Figure 10 biomimetics-08-00259-f010:**
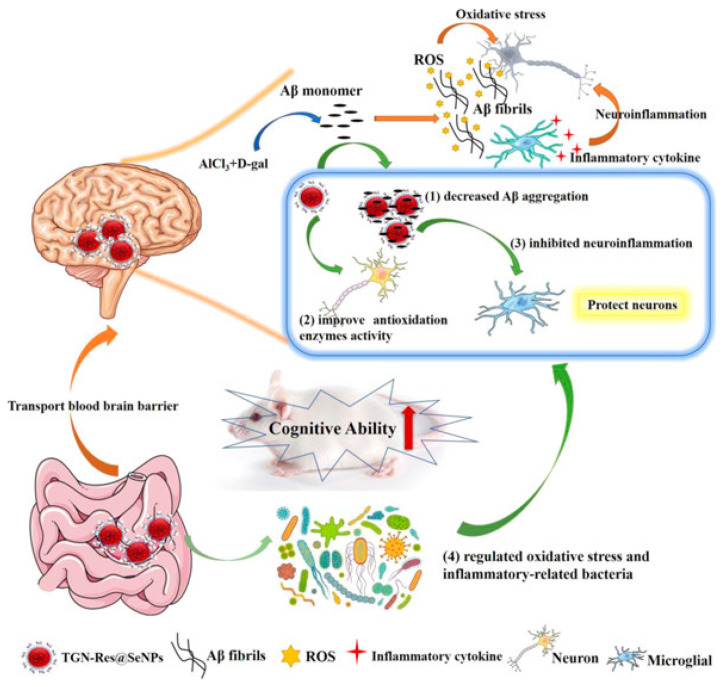
Schematic diagram of TGN-Res@SeNPs nanocomposites for the treatment of AD. Reprinted with permission from [84]. Copyright 2021, American Chemical Society.

**Figure 11 biomimetics-08-00259-f011:**
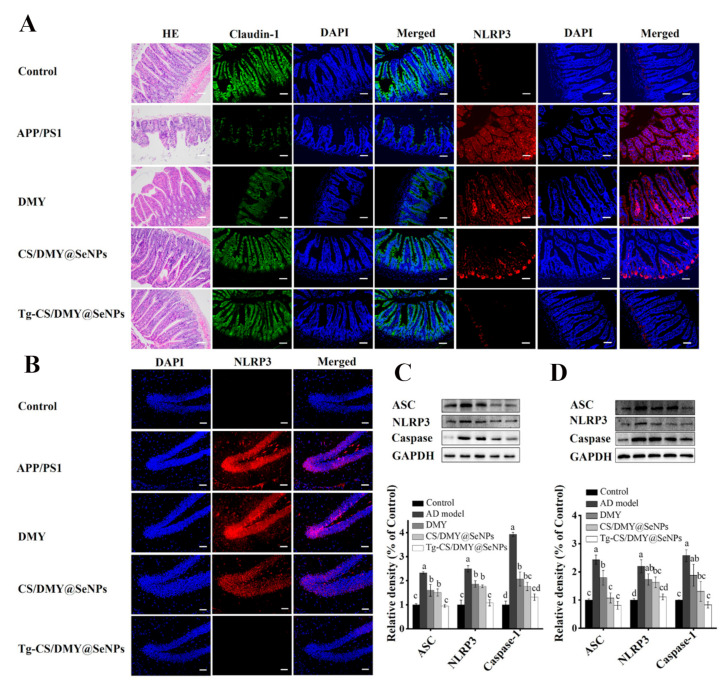
Effects of different surface modifications of SeNPs on the gut barrier permeability and protein expression level of NLRP3 in APP/PS1 mice. (**A**) HE staining, claudin-1 (green), and NLRP3 (red) immunofluorescence staining in mouse intestinal tissue. (**B**) NLRP3 immunofluorescence staining in brain tissue. (**C**) The protein expression level of ASC, nuclear NF-κB, and caspase-1 in the brain of mice. (**D**) The protein expression level of ASC, nuclear NF-κB, and caspase-1 in the intestine of mice. a–d represent significant differences among groups. Reprinted with permission from [87]. Copyright 2022, American Chemical Society.

**Figure 12 biomimetics-08-00259-f012:**
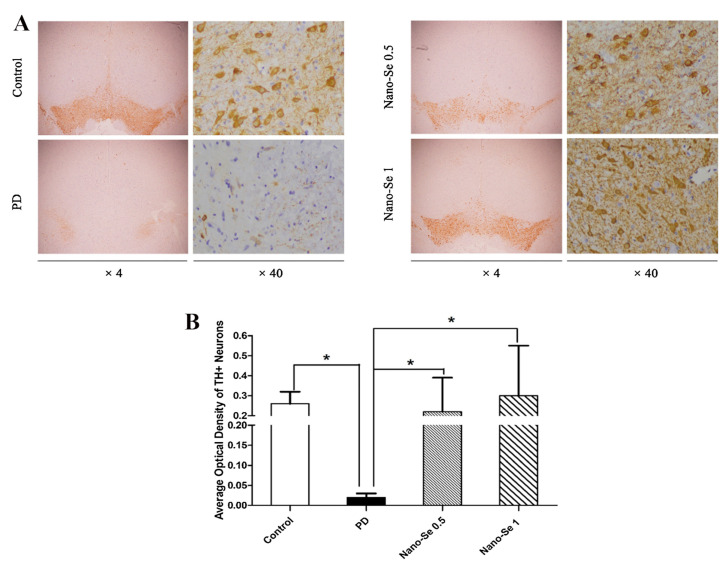
Gly@SeNPs can prevent the loss of dopaminergic neurons in rat model of PD. (**A**) Tyrosine hydroxylase (TH) immunohistochemistry of substantia nigra pars compacta (SNpc) sections from rats in the control group, PD model group, 0.05 mg/kg of glycine nano-selenium and 0.1 mg/kg of glycine nano-selenium. (**B**) The number of TH + neurons were significantly lower in the PD model group compared to the control group (* *p* < 0.05). Treatment with glycine nano-selenium significantly increased the number of TH + neurons as compared to the PD model group (all * *p* < 0.05). Reprinted from [91] with permission from Elsevier.

**Figure 13 biomimetics-08-00259-f013:**
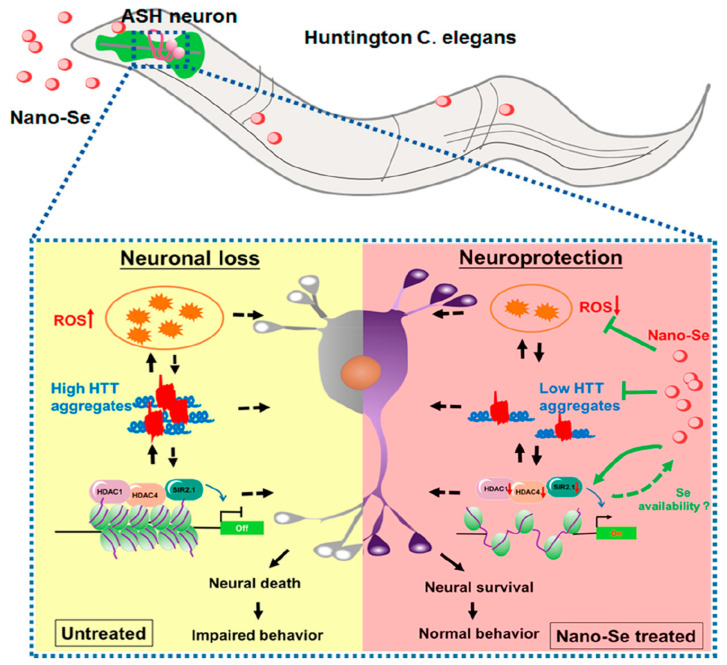
Schematic diagram of neuroprotective function of SeNPs in the Caenorhabditis elegans neurodegenerative HD model. Reprinted with permission from [94]. Copyright 2019, American Chemical Society.

**Figure 14 biomimetics-08-00259-f014:**
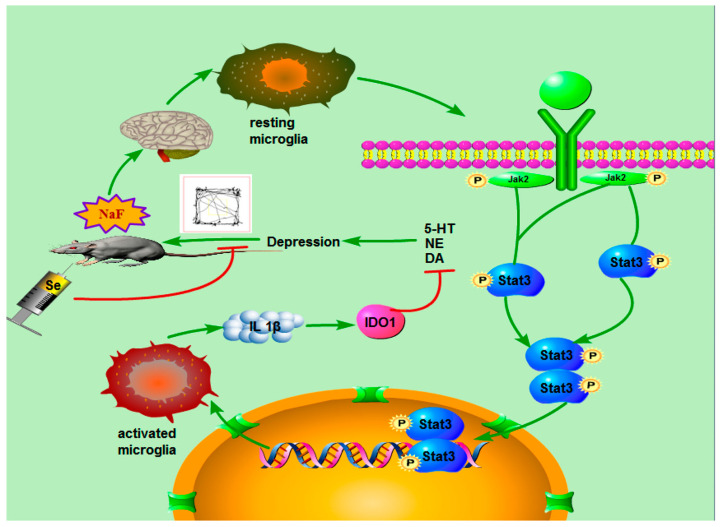
Molecular mechanism diagram of the protective effect of SeNPs on fluoride-induced depression behavior. Reprinted with permission from [98]. Copyright 2022, American Chemical Society.

**Figure 15 biomimetics-08-00259-f015:**
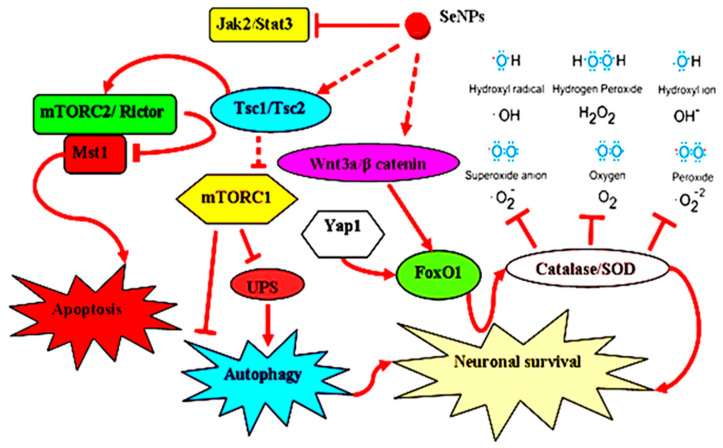
Different cellular signaling pathways that SeNPs may participate in stroke treatment. Reprinted with permission from [100] published under the Creative Commons Attribution license (CC BY 4.0) [66].

**Figure 16 biomimetics-08-00259-f016:**
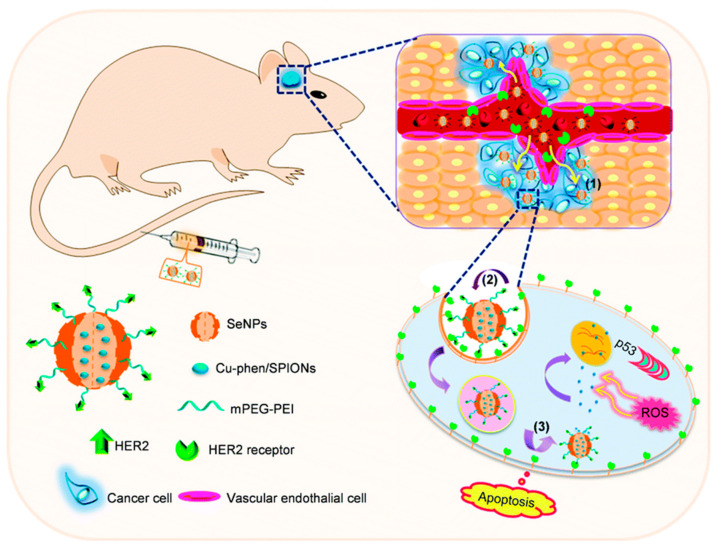
Schematic diagram of HER2@SeNPs crossing the BBB and penetrating into the U251 tumor sphere to kill U251 cells [102]. Used with permission of [Royal Society of Chemistry], from [Overcoming blood–brain barrier by HER2-targeted nano-system to suppress glioblastoma cell migration, invasion and tumor growth, Song, Zhenhuan; Liu, Ting; Chen, Tianfeng, 6, 4 and 2018]; permission conveyed through Copyright Clearance Center, Inc.

**Table 1 biomimetics-08-00259-t001:** Synthesis of selenium nanoparticles.

Synthesis Method		Modification Groups	Biomedical Applications	References
Physical synthesis		Active amino group	Antibacterial, etc.	Menazea et al. [17]
Chemical synthesis	Surfactants	PVA, PEG, etc.	Tissue regeneration, etc.	Cao et al. [18]
	Surface modification	Polyphenols (Active hydroxyl group, etc.)	Biological adhesion, Anti-inflammation, Antioxidation, etc.	Wang et al. [19] Yang et al. [20] Kumari et al. [21] Xu et al. [22]
		Polysaccharides (–OH, –COOH, –NH_2_ and –OSO_3_, etc.)	Antibacterial, Antioxidation, Drug delivery, Anticancer, etc.	Zou et al. [23] Dorazilova et al. [24] Zhai et al. [25] Rao et al. [26] Chen et al. [27] Zhou et al. [28] Tang et al. [29]
		Protein	Targeted therapy, Promote osteogenesis, Anticancer, etc.	Deng et al. [30] Zhuang et al. [31] Zhang et al. [32] Yu et al. [33]
		Polypeptide	Anti-inflammation, Targeted transport, etc.	Jiang et al. [34] Liu et al. [35]
		Drugs	Drug delivery, Cancer treatment, etc.	Deng et al. [36] Xia et al. [37]
		Genes	Gene delivery, Anticancer, etc.	Xia et al. [38]
Biosynthesis		None	Anti-diabetic oxidative stress, etc.	Fan et al. [11]

## Data Availability

Data derived from public domain resources.

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
