# Peer review of "The Developments of Surface-Functionalized Selenium Nanoparticles and Their Applications in Brain Diseases Therapy"

_biomimetics, 2023, doi:10.3390/biomimetics8020259_

Round 1

Reviewer 1 Report

The paper is very well written. I did not find much that is worth mentioning. Please check the English again and the quality and settings of the figures. Some addition references would be good.

Please check the English again .

Author Response

Response: We thank the Reviewer for his/her appreciation of our work, and we have imporved the Englisd and the quality and settings of the figures and also cited more specfic references in the revised manuscript according to the reviewers’ comments.

Reviewer 2 Report

Compared with inorganic and organic selenium, SeNPs are characterized by high stability, high bioavailability and low toxicity, meanwhile SeNPs can be easily modified by other functional groups to equip with more functions, which have more excellent therapeutic effects. The article introduces the synthesis (physical, chemical, biological), surface functionalization modification (polysaccharides, polyesters, proteins, etc.) and therapeutic effects of SeNPs in brain diseases (neurodegenerative diseases, brain glioma) in detail, with abundant information and clear hierarchy, which is a useful reference for the readers to understand this field. It is interesting, but there are major issues that must be addressed.

1.        According to the text, the article reviews the applications of SeNPs related to the treatment of brain diseases, but the abstract is limited to neurodegenerative diseases, which should be corrected. Besides that, the content about polyphenol groups and polysaccharide groups are similar in proportion in the article, so it is suggested to add polysaccharide groups in the keywords.

2.        The structure of the first paragraph of “introduction” is not reasonable, and it is suggested to describe in the order of overview-inorganic selenium-organic selenium. In addition, this part lacks relevant expressions on the physiological functions of selenium, which should be added.

3.        SeNPs exert therapeutic effects mainly through their anti-oxidative stress ability, while ROS are the main substances that cause oxidative stress, and excessive ROS lead to the development of many diseases in the organism. What are the main types of ROS? What are the enzyme systems that maintain redox homeostasis in the body? A separate paragraph should be added to discuss relevant issues. The following relevant literature can be consulted:

https://doi.org/10.1016/j.bioactmat.2022.08.022;

https://doi.org/10.1016/j.mtbio.2022.100215;

https://doi.org/10.1016/j.bioactmat.2021.06.006.

4.        References 9-12 cited in Figure2 are almost irrelevant to SeNPs, please determine if they are correctly cited and correct correspondingly. In addition, examples of the application of SeNPs in diabetes treatment should also be added.

5.        2.1 Physical synthesis: "Commonly used physical methods for the synthesis of SeNPs include hydrothermal treatments, microwave irradiation, and laser ablation" cites Ref. 14, which describes three methods of physical synthesis, but the following Figure3 cites Ref. 19, please explain the relationship between them.   

6.        2.1 Biosynthesis: " Biosynthesis, also known as green synthesis, rely on plants, fungi, and bacteria to synthesize SeNPs", but only plants and bacteria are mentioned later, and the examples related to fungi should be added.

7.        The review summarizes the current synthesis methods of SeNPs from physical, chemical and biological aspects and gives representative cases in each part with a clear structure. However, a horizontal comparison of the three synthetic methods is necessary, and a discussion of the advantages and disadvantages of each may make the article more informative.  

8.        In Figure8/Ref. 38, SeNPs are functionalized with RGDfC first, and then this complex function as a siSox2 carrier. In "Surface functionalization of selenium nanoparticles", the role of RGDfC should not be ignored.

9.        Is the main pathogenesis of neurodegenerative diseases the accumulation of free radicals and oxidative stress damage? Please explain why SeNPs can have a therapeutic effect by their anti-oxidative activity. A separate paragraph to elucidate the common pathogenesis of brain diseases may better enhance the coherence of the article and highlight the therapeutic role of SeNPs. 

10.    The review is well researched and provides many examples in terms of synthesis, surface modification, and biomedical applications of SeNPs. In order to make the structure of the article clearer, it is suggested that comprehensive tables be added to each of the three sections, to provide a more intuitive and comprehensive understanding of the synthesis techniques, a clear description of the roles and applications of the different types of modifiers, and the forms and functions of SeNPs used in different brain diseases.       

11.    Are references to other people's images authorized? If authorization has been obtained, it should be stated in the figure notes. In addition, Figure 5 is incomplete, and Figure 15 is unclear, so attention should be paid to the quality of the image.

12.    Abbreviations should be annotated with their full names the first time they appear. For example, Figure 7: “GPX2, GPX4...”, Figure 8: “RGDfC, siSox2...”, line 251: "EGCG", etc. After defining the abbreviations, such as “SeNPs”, the headings at all levels should be applied uniformly.  

13.    Line 299-302 missing key references.

Chapter 4. is rather vague in describing the prospects of SeNPs, and the outlook of "more functions" and "broader applications" of SeNPs can be expanded appropriately.     

 Moderate editing of English language

Reviewer 3 Report

The two corresponding authors of this review do not appear to be experts in the field: in the 85 cited references, not a single paper was authored by them. The result cannot but be a rather confused and messy manuscript. Writing a review is not just putting together the abstract of the collected paper (85 in this case) but requires their critical evaluation and the ability to precisely describe the different contributions of the systems reported. Regrettably, this has not been done here. Accordingly, the manuscript is not suitable for publication in its present form.

Specific observations:

1. Apparently SeNPs are involved in several redox processes. One would expect a description of the reasons why this happens. In Fig. 1 we find Se described in different oxidation states, but no comments have been made in the text in this regard.

2.  The preparation of the nanoparticles is mixed up with some properties of the nanosystems. This is logically not acceptable.

3. The intrinsic properties of SeNPs and those derived from their functionalization are not properly differentiated and one wonders wherefrom the described effects derive. The question remains mostly unanswered.

4. Also there is no clear differentiation between the passivation of the nanoparticles with inert agents and active ones. This adds up to the problem outlined in 3. Above.

5. In some cases acronyms are not defined—an example: EGCG.

6. In the Summary the authors state: “SeNPs with controllable size, high activity, low toxicity, and easy chemical modification shows high uptake efficiency, conversion efficiency, and better therapeutic effects in the treatment of neurological diseases”. Regrettably, aspects like the connection between size and effect, what affects the toxicity of the nanoparticles et cetera have not been adequately discussed. 

Many grammar mistakes are present

Round 2

Reviewer 3 Report

In this revised version of the manuscript, the authors have made a considerable effort in addressing my comments. No doubt: its quality increased significantly. Nevertheless, some adjustments are, in my opinion, still necessary.

1. p.3, lines 30-31 “[…] then transform Se(IV) and Se(VI) into either organic Se or SeNPs.” Organic Se and SeNPs do not represent oxidation states. What are the oxidation states of organic Se and SeNPs? Please state it here.

2. p. 3 line 36:  “[…] the toxicity of organic Se remains a major concern for in vivo applications.” What is the limit of toxicity? Please state it clearly.

3. Paragraph 3 deals with the functionalization of the nanoparticles. The authors never state what sort of interactions are present between the nanoparticles and the functionalizing molecules. Please state it clearly at the beginning of the paragraph.

4. Paragraphs 3.1 and 3.2 deal with polyphenols and polysaccharides functionalization. The sentence put after these paragraphs (lines 223-227): “Biomolecules, such as polysaccharides and polyphenols, can be used to stabilize the SeNPs  due to abundant hydroxyl, carbonyl, and other functional groups to interact with SeNPs. However, it was complex and difficult to identify the single effects of the nanocomposites composed of SeNPs and these biomolecules, because biomolecules can also regulate the bioactivities of SeNPs.” appears on one side a repetition, on the other it seems contradictory on the bases of what reported in the previous paragraphs. Perhaps I did not understand, and a clarification is possibly needed.

5. The sentence “Compared to inorganic Se and organic Se, SeNPs have higher biosafety and bioavailability, and the functionalization of SeNPs can affect their toxicity, surface electrical properties, and  stability.” (lines 230-231) is not supported by any reference or explanation. What are the bases for those statements? A clarification is needed.

6. The synthesis of the nanoparticles is highlighted in paragraph 2 (p. 6). Why is Table 1 reported on p. 28?

The English is rather poor throughout the manuscript.

See above

Author Response

Response to Reviewer 3 Comments

In this revised version of the manuscript, the authors have made a considerable effort in addressing my comments. No doubt: its quality increased significantly. Nevertheless, some adjustments are, in my opinion, still necessary.

Response: We thank the Reviewer for the appreciation and valuable suggestions of our revised work. We have studied comments carefully and have made corrections accordingly.

Point 1: p.3, lines 30-31 “[…] then transform Se(IV) and Se(VI) into either organic Se or SeNPs.” Organic Se and SeNPs do not represent oxidation states. What are the oxidation states of organic Se and SeNPs? Please state it here.

Response: Thanks for the insightful comments. We have re-written these sentences in the revised manuscript as follows.

“ The chemical cycle of selenium in living organisms indicated that plants prefer to absorb Se(IV) and Se(VI) in oxidation states due to their solubility (green arrows in the Figure 1). For organic selenium and selenium nanoparticles, they will be oxidized into Se(IV) or Se(VI) for absorption after they are entering the organism (solid orange arrows in Figure 1), and then converted into selenoproteins to participate in the metabolic activities of the organism.

Point 2. p. 3 line 36:  “[…] the toxicity of organic Se remains a major concern for in vivo applications.” What is the limit of toxicity? Please state it clearly.

Response: Thanks for the insightful comments. According to the reviewer’s suggestion, we have added several sentences to explain the toxicity of organic Se by citing some references. Please refer to the last paragraph on page 3 in the revised manuscript. 

Point 3. Paragraph 3 deals with the functionalization of the nanoparticles. The authors never state what sort of interactions are present between the nanoparticles and the functionalizing molecules. Please state it clearly at the beginning of the paragraph.

Response: Thanks for the insightful comments. We have added several sentences in the section 3 to describe the interactions between the nanoparticles and the functionalizing molecules.

“The SeNPs can be functionalized by integrating a variety of drugs or biomacromolecules based on the covalent or non-covalent coupling, such as the chemical conjugation of biomacromolecules on SeNPs and the physical adsorption onto the surface SeNPs”.

Please refer to Section 3 on Page 11 in the revised manuscript. 

Point 4. Paragraphs 3.1 and 3.2 deal with polyphenols and polysaccharides functionalization. The sentence put after these paragraphs (lines 223-227): “Biomolecules, such as polysaccharides and polyphenols, can be used to stabilize the SeNPs due to abundant hydroxyl, carbonyl, and other functional groups to interact with SeNPs. However, it was complex and difficult to identify the single effects of the nanocomposites composed of SeNPs and these biomolecules, because biomolecules can also regulate the bioactivities of SeNPs.” appears on one side a repetition, on the other it seems contradictory on the bases of what reported in the previous paragraphs. Perhaps I did not understand, and a clarification is possibly needed.

Response: Thanks for the insightful comments. To avoid the misleading, we have re-organized this paragraph in the revised manuscript as follows. 

“In short, biomolecules, such as polysaccharides and polyphenols, can be used to not only stabilize the SeNPs due to abundant hydroxyl, carbonyl, and other functional groups to interact with SeNPs, but also enhance the bioactivities of functionalized SeNPs with the synergistic effects derived from biomolecules themselves”.

Point 5. The sentence “Compared to inorganic Se and organic Se, SeNPs have higher biosafety and bioavailability, and the functionalization of SeNPs can affect their toxicity, surface electrical properties, and stability.” (lines 230-231) is not supported by any reference or explanation. What are the bases for those statements? A clarification is needed.

Response: We thank the reviewer for careful reading of the manuscript. We have made changes and added a reference to support the statement in the revised manuscript, please refer to literature 8 on page 16.

Point 6. The synthesis of the nanoparticles is highlighted in paragraph 2 (p. 6). Why is Table 1 reported on p. 28?

Response: We thank the reviewer for their careful reading of the manuscript. According to reviewer’s consideration, we have changed Table 1 on page 7 in the revised manuscript.

Point 7. The English is rather poor throughout the manuscript.

Response: We have modified the English throughout the manuscript.